

# Hallmarks of glycogene expression and glycosylation pathways in squamous and adenocarcinoma cervical cancer

Patricia Martinez-Morales[1], Irene Morán Cruz[2], Lorena Roa-de la Cruz[3], Paola Maycotte[4], Juan Salvador Reyes Salinas[5], Victor Javier Vazquez Zamora[5], Claudia Teresita Gutierrez Quiroz[5], Alvaro Jose Montiel-Jarquin[5] and Verónica Vallejo-Ruiz[2]

[1] CONACYT-Centro de Investigación Biomédica de Oriente, Mexican Institute of Social Security, Metepec, Puebla, México
[2] Centro de Investigación Biomédica de Oriente, Laboratory of Molecular Biology, Instituto Mexicano del Seguro Social, Metepec, Puebla, México
[3] Department of Biological Chemical Sciences, Universidad de las Américas-Puebla, San Andrés Cholula, Puebla, Mexico
[4] Centro de Investigación Biomédica de Oriente, Laboratory of Cell Biology, Instituto Mexicano del Seguro Social, Metepec, Puebla, México
[5] Hospital de especialidades, General Manuel Ávila Camacho, Instituto Mexicano del Seguro Social, Puebla, Puebla, México

Corresponding author
Verónica Vallejo-Ruiz,
veronica.vallejor@imss.gob.mx

## ABSTRACT

**Background:** Dysregulation of glycogene expression in cancer can lead to aberrant glycan expression, which can promote tumorigenesis. Cervical cancer (CC) displays an increased expression of glycogenes involved in sialylation and sialylated glycans. Here, we show a comprehensive analysis of glycogene expression in CC to identify glycogene expression signatures and the possible glycosylation pathways altered.

**Methods:** First, we performed a microarray expression assay to compare glycogene expression changes between normal and cervical cancer tissues. Second, we used 401 glycogenes to analyze glycogene expression in adenocarcinoma and squamous carcinoma from RNA-seq data at the cBioPortal for Cancer Genomics.

**Results:** The analysis of the microarray expression assay indicated that CC displayed an increase in glycogenes related to GPI-anchored biosynthesis and a decrease in genes associated with chondroitin and dermatan sulfate with respect to normal tissue. Also, the glycogene analysis of CC samples by the RNA-seq showed that the glycogenes involved in the chondroitin and dermatan sulfate pathway were downregulated. Interestingly the adenocarcinoma tumors displayed a unique glycogene expression signature compared to squamous cancer that shows heterogeneous glycogene expression divided into six types. Squamous carcinoma type 5 (SCC-5) showed increased expression of genes implicated in keratan and heparan sulfate synthesis, glycosaminoglycan degradation, ganglio, and globo glycosphingolipid synthesis was related to poorly differentiated tumors and poor survival. Squamous carcinoma type 6 (SCC-6) displayed an increased expression of genes involved in chondroitin/dermatan sulfate synthesis and lacto and neolacto glycosphingolipid synthesis and was associated with nonkeratinizing squamous cancer and good survival. In summary, our study showed that CC tumors are not a

uniform entity, and their glycome signatures could be related to different clinicopathological characteristics.

## INTRODUCTION

Cervical cancer (CC) is one of the leading causes of cancer-related death in women worldwide (*International Agency for Research on Cancer (IARC), 2020*). According to the histological origin, the most frequent types of CC are squamous cell carcinoma (SCC) and adenocarcinoma (AC) in 80% and 15–20% of cases, respectively (*Chavarro et al., 2009*). A recent multi-omic molecular characterization of CC leads to the molecular classification into keratin-low squamous, keratin-high squamous, and adenocarcinoma-rich cancer (*The Cancer Genome Atlas Research Networkk (TCGA), 2017*), suggesting that CC is not a uniform entity. In this context, the glycome of CC may also be diverse.

Glycogenes, or genes associated with glycan synthesis, include those genes implicated in the glycosylation pathways, such as glycosyltransferases and glycosidases, and genes necessary for glycosynthesis, such as sugar-nucleotide synthases, sugar-nucleotide transporters, chaperones, genes necessary for glycolysis (*Sachiko & Kiyohiko, 2019*; *Schjoldager et al., 2020*). Several studies support that cancer displays an aberrant expression of those glycogenes implicated in glycosylation (*Pinho & Reis, 2015*; *Schjoldager et al., 2020*); and certainly, changes in the expression of glycogenes or cell surface glycans have been related to cancer progression and prognosis (*Kannagi et al., 2008*; *Pinho et al., 2012*; *Pinho & Reis, 2015*; *Munkley & Elliott, 2016*; *Zhuo, Li & Guan, 2018*; *Schjoldager et al., 2020*; *Yip, Smollich & Götte, 2006*; *Liu et al., 2010*; *Cordeiro Pedrosa et al., 2015*; *Li et al., 2017*; *Lin et al., 2018*; *Salustiano et al., 2020*). In this context, *in silico* models predicting cellular glycosylation are important for addressing the cancer glycome (*Schjoldager et al., 2020*).

Several reports indicate that cervical dysplasia and CC display increased expression of the glycan-cell surface sialyl Lewis A (SLeA) and sialyl Lewis X (SLeX) (*Roy & Chakraborty, 2005*; *López-Morales et al., 2010*; *Engelstaedter et al., 2012*; *Velázquez-Márquez et al., 2012*; *Jin et al., 2016*) and polylactosamine and the O-glycosylated antigens Tn and sialyl-Tn (*Numa et al., 1995*; *Clark et al., 2014*). Even the expression of the Tn antigen correlates with metastatic potential and poor prognoses (*Numa et al., 1995*). In concordance, the expression of sialyltransferases ST3Gal III and ST6Gal I is increased in premalignant lesions and CC (*Wang et al., 2001*; *Wang et al., 2002*; *López-Morales et al., 2009*). Even a high expression of the glycogene *ST6GAL1* correlates with stromal invasion, metastatic spread to the lymph nodes, and poor patient prognosis (*Wang et al., 2002*; *Wang et al., 2003*). Concerning polylactosamine synthesis, the expression of the polylactosamine synthase β3GnT2 is increased in high-grade premalignant lesions

(*Clark et al., 2014*), while the expression of the glycogene *B4GALT3* is upregulated in CC (*Sun et al., 2016*). Other changes include the decrease of fucosylation level as the grade of cervical dysplasia increases (*Jin et al., 2016*). Here, to obtain comprehensive scenery of the CC glycome and a glycogene signature related to the clinicopathological characteristics and prognosis, we first compared the glycogene expression in CC with the normal cervix and then between the histological types of AC and SCC.

## MATERIALS & METHODS

### Patient samples

The Ethics Committee approved the present study of the Instituto Mexicano del Seguro Social (R-2012-785-061). All patients signed an informed consent form according to the guidelines of the Human Ethics Committee. Biopsies were obtained at the Hospital de Especialidades, General Manuel Ávila Camacho, Instituto Mexicano del Seguro Social from women between the ages of 20 and 73. For the CC group, inclusion criteria included adult women with SC diagnosis with no previous treatment (mean age 42). In contrast, for the normal cervix group, samples were obtained from adult women with uterine myomatosis diagnosis (mean age 35). A total of 23 CC samples and 15 biopsies of normal cervical tissue were included in the analysis.

### Microarray assay

Normal cervix tissues and CC samples were collected and maintained in RNAlater solution (Qiagen, Hilden, Germany) at −80 °C. Total RNA was extracted using the RNeasy Plus system (Qiagen, Hilden, Germany) following the manufacturer's protocol. RNA samples were pooled into two groups (normal and CC). Microarray analysis was performed at the Microarray Unit of the Instituto de Fisiología Celular, Universidad Nacional Autónoma de México (UNAM). A total of ten micrograms of total RNA were used to synthesize labeled cDNA employing the First-Strand cDNA labeling kit (Invitrogen, Waltham, MA, USA), incorporating dUTP-Alexa555 or dUTP-Alexa647. Equal quantities of labeled cDNA were hybridized using IniHyb hybridization solution (TeleChem International INC., Los Altos, CA, USA). The microarray included 10,000 genes from the NCB and GenBank databases. Acquisition and quantification of array images were performed using a Scan Array 4,000 instrument and ScanArray 4,000 software (Packard BioChips,Waltham, MA, USA). The Alexa555 and Alexa647 density and background mean values were calculated with ArrayPro Analyzer software from Media Cybernetics. GenArise1.38.0 software developed in the Computing Unit of the Cellular Physiology Institute of UNAM was used for the data analysis and statistics. Analyzed data were submitted to the NCBI-Gene Expression Omnibus database (accession number GSE159976).

### Glycogene expression analysis

We selected glycogenes with a z-score ≥ 2 to identify the glycogenes with expression changes between the CC and normal cervix. A total of 401 glycogenes reported to date were used for glycogene expression analysis; these were obtained from the published reports of *Venkitachalam et al. (2016)*, and *Aco-Tlachi et al. (2018)*. Glycogenes included

glycosyltransferases and glycosidases, sugar-nucleotide synthases, sugar-nucleotide transporters, chaperones, and some genes related with energetic metabolism.

## Protein-protein enrichment analysis

Glycogenes identified as altered were submitted to the STRING database version 10.5 (http://string-db.org/) to evaluate predicted protein-protein interactions (Szklarczyk et al., 2019). We considered the following settings for the analysis: text mining, experiments, databases, coexpression, neighborhood, gene fusion, and cooccurrence as interaction sources, with no more than five interactors and a minimum interaction score of 0.4 as the confidence level. Biological process (GO) and KEGG pathways were chosen as functional enrichments of the network. The results exclusively included a protein-protein enrichment $p$-value of at least $\leq 0.05$ and an FDR of at least $\leq 0.05$.

## RNA-seq data analysis

RNA-seq data of 401 glycogenes from TCGA and Firehose Legacy were obtained from the cBioPortal for Cancer Genomics (Gao et al., 2013). We analyzed glycogene expression considering the mRNA expression of genes with a z-score threshold $\geq 2.0$ relative to all samples (log RNA Seq V2 RSEM). The population included women from 20 to 88 with a diagnosis of SCC and AC, four samples of cervical adenosquamous carcinoma, and all the group stages from I to IVB (AJCC staging system).

## Hierarchical clustering

Clinical data of patients obtained from the cBioPortal for Cancer Genomics and their respective RNA-seq data of glycogene expression were organized and submitted in the online free software Morpheus (https://software.broadinstitute.org/morpheus/) to perform the heat map analysis and hierarchical clustering. One minus Pearson correlation was used as the metric, and the average was used as the linkage method. Venn diagram of glycogenes clusters were obtained using the tool http://bioinformatics.psb.ugent.be/webtools/Venn/.

## Pathway enrichment analysis

Altered glycogenes were analyzed using DAVID (Database for Annotation, Visualization, and Integrated Discovery) software (http://david.abcc.ncifcrf.gov/). Based on gene ontology (GO), genes were classified according to their function in the KEGG pathway with at least a $p$-value < 0.05 and FDR < 0.05.

# RESULTS

## Comparison of glycogene expression between cervical cancer and healthy cervical tissue

We performed a microarray expression assay to identify changes in glycogene expression in CC compared to a normal cervix. From the 178 glycogenes contained in the microarray, results indicated that 15 glycogenes were altered in CC (z-score threshold $\geq 2.0$). The glycogenes *PIGC, PIGN, SMPD3, CHI3L1, HEXB,* and *ST6GAL1* were upregulated (Table S1-column A), and *GYS1, CHST12, CTBS, HAS3, HPSE, HYAL2, GALNT11,*

**Table 1 Glycogenes increased in cervical cancer patients and aberrantly expressed in several types of cancers.**

| Glycogene | Enzyme function | Reports in cancer patients |
|---|---|---|
| *PIGC* | Encodes for the phosphatidylinositol N-acetylglucosaminyltransferase subunit C that is involved in the glycosilphosphatidylinol anchor biosynthesis | No reported as altered in cancer patients |
| *PIGN* | Encodes for GPI ethanolamine phosphate transferase 1 that is involved in the glycosilphosphatidylinol anchor biosynthesis | Expression aberration is associated with progression in acute myeloid leukemia (*Teye et al., 2017*) |
| *SMPD3* | Sphingomyelin phosphodiesterase 3 that catalyzes the hydrolysis of sphingomyelin to form ceramide and phosphocholine | Associated with good prognosis in gastric cancer (*Liu et al., 2018*) |
| *CHI3L1* | Chitinase-3-like protein 1. Although it belongs to the glycosyl hydrolase 18 family, Leu-140 is present instead of the conserved Glu which is an active site residue. Therefore, this protein lacks chitinase activity. | Expression associated with vasculogenic mimicry in cervical cancer patients (*Ngernyuang et al., 2018*) and its high expression is associated with poor outcome and chemoresistance in ovarian cancer patients (*Lin et al., 2019*) |
| *HEXB* | Beta-hexosaminidase subunit beta involved in hydrolysis of gangliosides GM2 to GM3 | Upregulation in invasive ductal carcinoma-associated blood vessels (*Jones et al., 2012*) and poor survival of melanoma patients (*Welinder et al., 2017*) |
| *ST6GAL1* | Beta-galactoside alpha-2,6-sialyltransferase 1 that transfers sialic acid from CMP-sialic acid to galactose-containing acceptor substrates | Upregulation in pancreatic, prostate, breast and ovarian cancer (*Garnham et al., 2019*). In cervical cancer is increased in metastatic cancer and its levels correlate with stromal invasion, metastatic spread to the lymph nodes and poor patient prognosis (*Wang et al., 2002*; *Wang et al., 2003*) |

*UGT2B4*, and *UGT2B28* were downregulated (Table S2-column A). Bibliographic research indicates that most altered glycogenes are reported to have expression changes in other types of cancer (Tables 1 and 2).

Then, to further map the 15 altered genes onto the global glycogene network, we performed a protein-protein interaction (PPI) network functional analysis in the STRING database (*Szklarczyk et al., 2019*) using the 401 glycogenes (Fig. S1A). Results showed a complex interaction among the 401 glycogenes, since some of them were involved in several glycosylation pathways (Figs. S1B, S1C, S1F). Thus, we refine the search for identification of specific glycosylation pathways by performing a PPI analysis, including only the upregulated and downregulated glycogenes. The glycogenes *SMPD3*, *CHI3L1*, *HEXB*, and *ST6GAL1*, were involved in the immune system response ($p$-value $\leq 0.005$), while *PIGC* and *PIGN* participate in the synthesis of GPI anchors (Fig. 1A). In contrast, results revealed that *HYAL2* and *HAS3* were involved in the hyaluronan (HA) metabolic process. To further identify putative targets and additional glycosylation pathways not displayed in the previous analysis, we included the upregulated or downregulated glycogenes and five more interactors in the subsequent analysis. The results again showed that *PIGC* and *PIGN* were involved in synthesizing GPI anchors (Fig. 1C). In contrast, *CHI3L1* and *HEXB*, together with *FUCA2* and *HEXA*, were engaged in neutrophil degranulation ($p$-value $\leq 0.001$) (Fig. 1C). Concerning the downregulated glycogenes, *UGT2B4* and *UGT2B28*, along with *UGP2*, were implicated in pentose and glucuronate interconversions ($p$-value $\leq 0.001$) (Fig. 1D). Other results showed that *CHST12* could interact with *CHST3* and *CHST7* in chondroitin sulfate (CS)/dermatan sulfate synthesis ($p$-value $\leq 0.001$) (Fig. 1D).

**Table 2 Glycogenes downregulated in cervical cancer patients and aberrantly expressed in several types of cancers.**

| Glycogene | Enzyme function | Reports in cancer patients |
|---|---|---|
| GYS1 | Glycogen synthase that transfers the glycosyl residue from UDP-Glc to the non-reducing end of alpha-1,4-glucan. | Overexpression of GYS1 along with MIF is associated with adverse outcome in acute myeloid leukaemia (*Falantes et al., 2015*) |
| CHST12 | Carbohydrate sulfotransferase 12 that transfers sulfate to position 4 of the N-acetylgalactosamine (GalNAc) residue of chondroitin and desulfated dermatan sulfate | High expression of mRNA in ovarian cancer (*Oliveira-Ferrer et al., 2015*) |
| CTBS | Hydrolyze of N-acetyl-beta-D-glucosamine (1–4)N-acetylglucosamine chitobiose core from the reducing end of the bond. | No reported as altered in cancer patients |
| HAS3 | Hyaluronan synthase 3 catalyzes the addition of GlcNAc or GlcUA monosaccharides to the nascent hyaluronan polymer. | HAS3 underexpression is associated with poor prognosis in patients with urothelial carcinoma of the upper urinary tract and urinary bladder (*Chang et al., 2015*) |
| HPSE | Heparanase That cleaves heparan sulfate proteoglycans into heparan sulfate side chains and core proteoglycans | The positive expression is associated with prognosis in ovarian cancer (*Zhang et al., 2013*). Expression levels are associated with tumor size, histology grade and pathological stage in breast cancer (*Tang et al., 2014*). High levels of HPSE is associated with shorter survival of patients with high-grade glioma (*Kundu et al., 2016*) |
| HYAL2 | Hyaluronidase-2 that hydrolyzes high molecular weight hyaluronic acid to produce an intermediate-sized product | Expression of HYAL2 is negatively correlated with lymphatic metastasis and TNM stage in colorectal cancer (*Jin et al., 2019*). Combined expression of HYAL2 and S100P is associated with shorter progression-free survival, recurrence events, and occurrence of metastasis (*Maierthaler et al., 2015*). HYAL2 along with other four genes discriminated progressive from non-progressive bladder cancer patients (*van der Heijden et al., 2016*). Reduced HYAL1 expression was associated with the progression of endometrial carcinomas and deep myometrial invasion (*Nykopp et al., 2015*) |
| GALNT11 | N-acetylgalactosaminyltransferase 11 that catalyzes the initiation of protein O-linked glycosylation | GALNT11 expression is associated disease prognosis (*Libisch et al., 2014*) |
| UGT2B4 | UDP-glucuronosyltransferase 2B4 that transfer glucoronate to an acceptor to produce an acceptor β-D-glucuronoside | Increased in hepatocellular carcinoma and colorectal cancer patients (*Kondoh et al., 1999*; *Sayagués et al., 2016*) |
| UGT2B28 | UDP-glucuronosyltransferase 2B28 that transfer glucoronate to an acceptor to produce an acceptor β-D-glucuronoside | Not reported as aberrantly expressed in cancer but genomic variation is associated with ages of hepatocellular carcinoma occurrence and life expectancy (*Le et al., 2019*) |

## Comparison between adenocarcinoma and squamous cervical

Next, to identify glycogene signatures in CC, we analyzed the expression of 401 glycogenes in 297 samples using the RNA-seq public data obtained from the cBioPortal for Cancer Genomics using the samples that corresponded to the histological types of SC and AC (Table 3). Unsupervised hierarchal clustering analysis revealed that 25 genes exhibited low expression in almost all samples compared with the rest of the glycogenes (Table S2-column B), suggesting a hallmark of glycogene expression among all CC types.

Then, to test whether AC and SCC could display distinctive glycogene expression, we compared the gene expression between them. The results showed differences in glycogene expression. Samples of SCC displayed a heterogeneous pattern of gene expression (Fig. 2A). In contrast, AC displayed characteristic glycogene expression that included 107 glycogenes grouped into three different clusters: AC-A, AC-B, and AC-C (Fig. 2A). Cluster AC-A included 51 genes with a high expression (Fig. 2A;

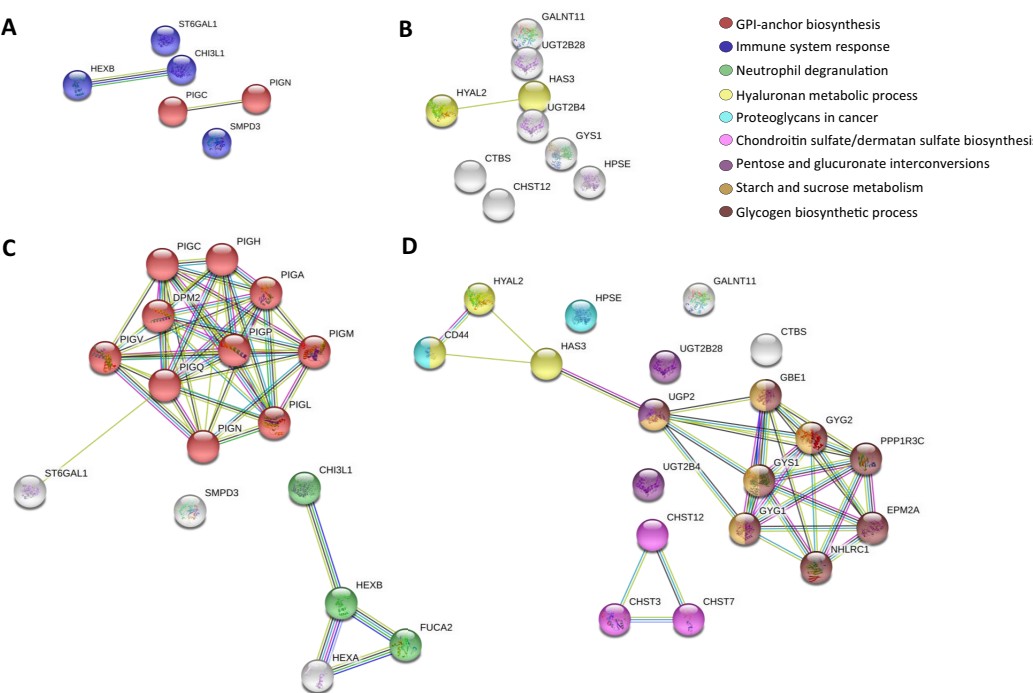

**Figure 1 STRING protein-protein interaction network among the altered glycogenes in cervical cancer in comparison with a normal cervix.** (A) Interaction among the upregulated glycogenes shows two predicted cellular processes: the glycosylphosphatidylinositol (GPI) biosynthesis process (in red) and an immune system response process (in blue). (B) Interaction among the downregulated glycogenes shows the predicted network between HYAL2 and HAS3 in the hyaluronan metabolic process (in yellow). (C) Interaction among the upregulated glycogenes with five predicted interactors shows that the glycogenes PIGC and PIGN interact with more glycogenes of the GPI biosynthesis process (in red), while CHI3L1 and HEXB are predicted to interact with FUCA2 in the neutrophil degradation process (in green). (D) Interaction among the downregulated glycogenes and their respective predicted interactors shows four networks: HYAL2 and HAS3 along with CD44 participate in the hyaluronan metabolic process (in yellow); also, HAS3 is predicted to interact with UGP2 in pentose and glucuronate interconversions (in purple), glycogen biosynthetic process (in brown) and starch and sucrose metabolism (in gold). In addition, the glycogene CHST12 was predicted to interact with CHST3 and CHST7 in the chondroitin sulfate/dermatan sulfate biosynthesis process (in violet)

**Table 3 Clinical characteristics of 297 cervical cancer patients evaluated by RNA-seq analysis.**

|  | n | Stage | | | | | Tumor differentiation grade | | | | | | Overall survival status | |
|---|---|---|---|---|---|---|---|---|---|---|---|---|---|---|
|  |  | I | II | III | IV | No data | G1 | G2 | G3 | G4 | GX | No data | Living | Deceased |
| Squamous Cell Carcinoma | 253 | 126 | 62 | 42 | 16 | 7 | 12 | 109 | 103 | 1 | 20 | 8 | 192 | 61 |
| Adenocarcinoma | 44 | 30 | 7 | 3 | 4 | – | 5 | 25 | 11 | – | 3 | – | 34 | 10 |

Table S1-column B), while AC-B and AC-C displayed low gene expression of 23 and 33 glycogenes (Table S2-column C and column D), respectively, compared with the rest of the CC population (Fig. 2A). The results of the functional analysis revealed that glycogenes in the AC-A cluster participate in N-glycan biosynthesis, mucin-type O-glycan

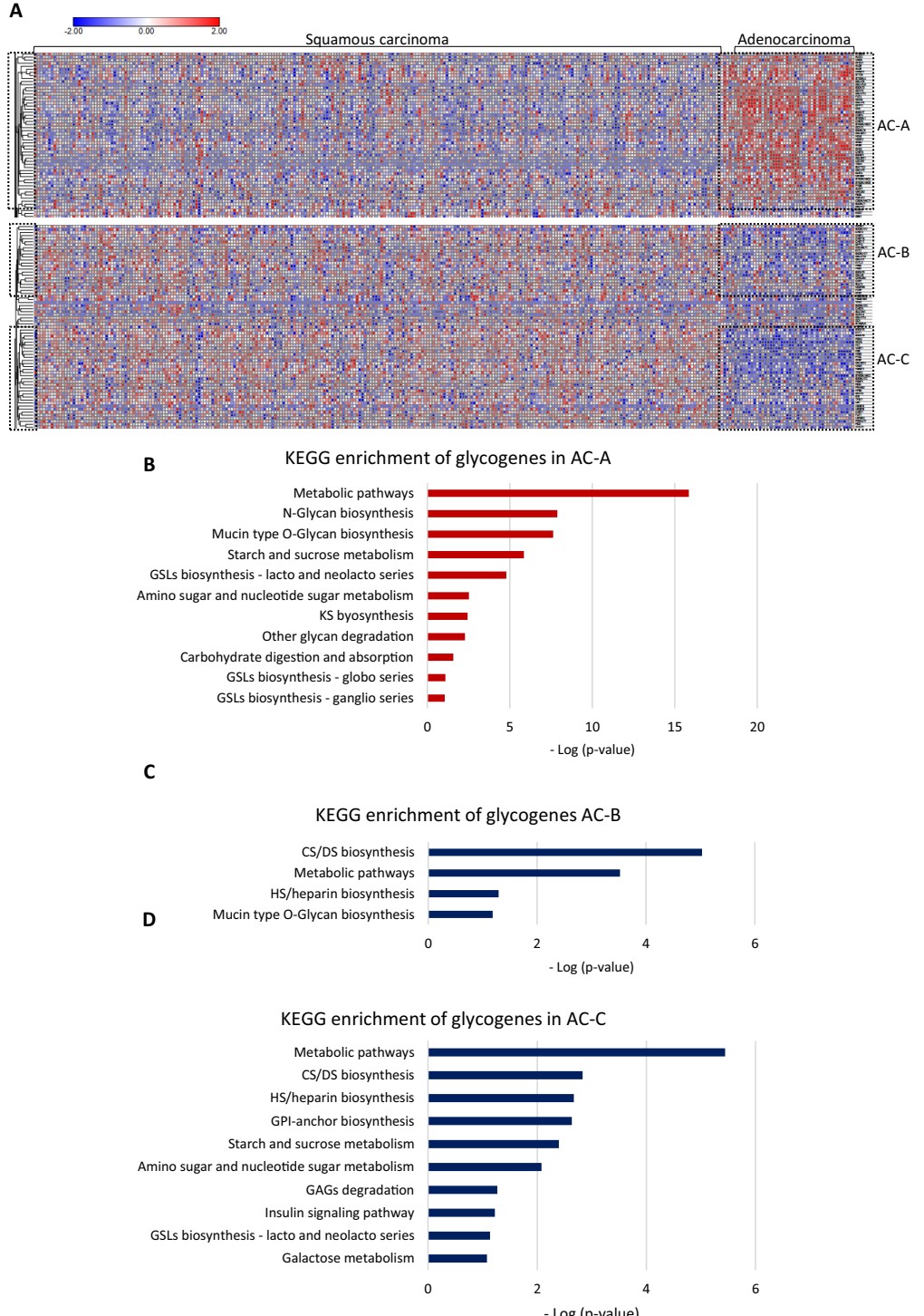

**Figure 2 Glycogene expression comparsion between adenocarcinoma and squamous carcinoma.** Glycogene expression comparsion between adenocarcinoma and squamous carcinoma. (A) Heat map of glycogene expression in squamous carcinoma and adenocarcinoma and clusters of glycogenes. (B) KEGG pathway enrichment analysis of glycogenes in AC-A, AC-B (C) and, AC-C (D). Red color indicates high expresión and blue color low expression. GSLs: glycosphincolipids; GAGs: glycosami-noglycans; GPI: glycosylphosphatidylinositol  

biosynthesis, glycosphingolipid biosynthesis (lacto and neolacto series), GAG biosynthesis keratan sulfate (KS), glycosphingolipid biosynthesis (globo and ganglio series), other glycan degradation, and other pathways, such as amino sugar and nucleotide sugar metabolism, starch and sucrose metabolism, and carbohydrate digestion and absorption (Fig. 2B). In comparison, genes with a low expression in AC-B are involved in GAG biosynthesis (CS, dermatan sulfate (DS)) and heparan sulfate/heparin (HS), mucin-type O-glycan biosynthesis, and metabolic pathways (Fig. 2C). Similarly, glycogenes in cluster AC-C were implicated in metabolic pathways and biosynthesis of CS, DS, HS but also GPI-anchor biosynthesis, glycosphingolipid biosynthesis (lacto and neolacto series), galactose metabolism, GAG degradation, and other pathways, such as amino sugar and nucleotide sugar metabolism, starch and sucrose metabolism, and insulin signaling pathways (Fig. 2D).

We further compared the glycogene expression of adenosquamous carcinoma with AC and SCC since the histological type includes both adenoid and squamous cells from the origin.

## Identification of hallmark glycogenes in squamous cervical cancer

To further identify a glycogene expression hallmark in SCC, we removed the 20 glycogenes with low expression in all the CC samples and performed an unsupervised hierarchal clustering analysis exclusively in SCC samples ($n$ = 253). The results indicate that almost all SCC samples exhibit a low expression of *CHIA, GLT6D1, SPACA3, FUT9, B3GALT1, GALNTL5, NEU2, UGT2B15, UGT2A3, UGT2B7,* and *LYZL4* (Table S2-column E). Moreover, the results showed that two clusters of samples, SCC-1 and SCC-2, display specific glycogene expression (Fig. 3A). SCC-1 comprised 29 samples that showed a cluster of 30 glycogenes with high expression (SCC-1A; Table S1-column C) and a cluster of 19 genes with low expression (SCC-1B) (Fig. 3 and Table S2-column F). In contrast, SCC-2 formed by 36 samples exhibited a cluster of 31 genes with low expression (Fig. 3A and Table S2-column G).

SCC-1 included samples with a diagnosis from stage I to stage III, but most of the samples were stage I, and patients displayed (Table 4) an 89% survival rate. Samples included moderately differentiated (G2) and poorly differentiated (G3) tumors (Table 4). Enrichment analysis showed that genes were impl1icated in N-glycan biosynthesis, other types of O-glycan biosynthesis, GPI-anchor biosynthesis, lysosomes, GAG degradation, biosynthesis of CS/DS and HS, and other processes, such as processes in the endoplasmic reticulum and amino sugar and nucleotide sugar metabolism (Fig. 3C). Glycogenes in cluster SCC-1B were involved in N-glycan biosynthesis, CHS/DS biosynthesis, sphingolipid metabolism, and protein processing in the endoplasmic reticulum (Fig. 3C).

Regarding cluster SCC-2 and the clinicopathological characteristics, most of the samples corresponded to stage I; however, two samples with a diagnosis of stage IV and comprised of differentiation grades from G1 (well-differentiated) to G3 were also included. Overall survival analysis showed that patients displayed an 88% survival rate (Table 4). The glycogenes were involved in CS/DS biosynthesis, GAG degradation, other glycan degradation, and glycosphingolipid biosynthesis (ganglio, globo, lacto and neolacto series),

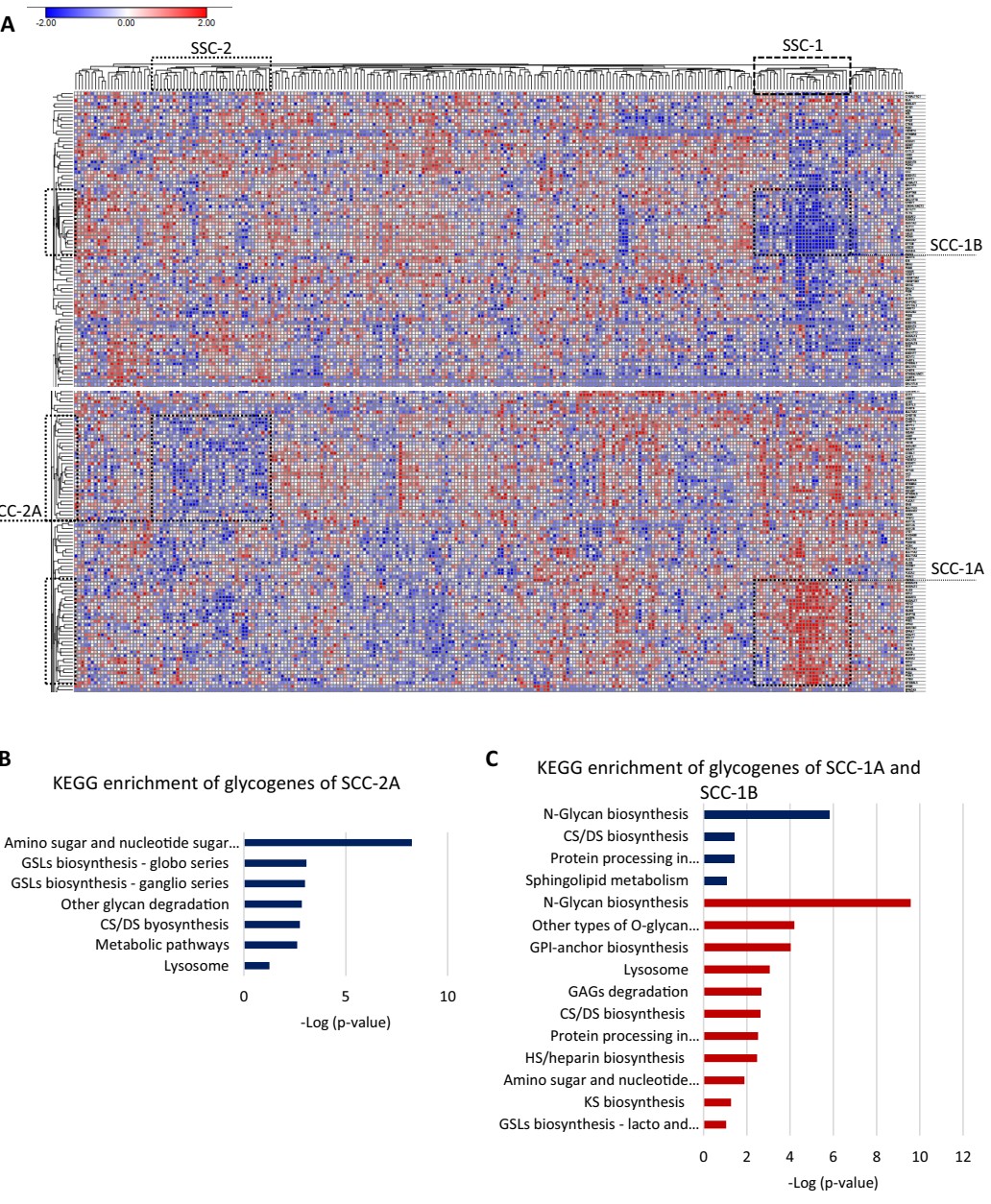

**Figure 3 Screening of the glycogene expression pattern in cervical squamous carcinoma.** (A) Heat map of glycogene expression and unsupervised hierarchal analysis showing two different clusters of patient samples, SCC-1 and SCC-2, that display certain clusters of glycogenes (SCC-1A and SCC-1B, and SCC-2A, respectively). (B) KEGG enrichment analysis of the cluster of glycogenes with low expression in SCC-2A. (C) KEGG enrichment analysis of glycogenes with high expression (SCC-1A) and low expression (SCC-1B). CS: chondroitin sulfate; DS dermatan sulfate; HS: heparan sulfate; GSLs: glyco-sphingolipids; GAGs: glycosaminoglycans; GPI: glycosylphosphatidylinositol. Red indicates high expression, while blue indicates low expression.

among others (Fig. 3C). These results indicate that at least two types of SCC samples display a hallmark of glycogene expression. However, we cannot conclude a specific association with tumor characteristics or patient survival because not all the samples contained complete information.

**Table 4 Clinical characteristics of the squamous cervical cancer types according to their glycogene expression.**

| | Molecular classification | | | Clinical stage classification | | | | | Neoplasm histologic grade | | | | | | Overall survival status | |
| --- | --- | --- | --- | --- | --- | --- | --- | --- | --- | --- | --- | --- | --- | --- | --- | --- |
| | Keratinizing carcinoma | Non-keratinizing carcinoma | No data | Stage I | Stage II | Stage III | Stage IV | No data | G1 | G2 | G3 | G4 | GX | No data | Alive | Deceased |
| SCC-1 (*n* = 29) | 7 | 15 | 7 | 16 | 7 | 5 | 0 | 2 | 0 | 16 | 13 | 0 | 0 | 0 | 26 | 3 |
| SCC-2 (*n* = 35) | 6 | 19 | 11 | 20 | 8 | 5 | 2 | 1 | 1 | 16 | 14 | 0 | 3 | 2 | 31 | 5 |
| SCC-3 (*n* = 25) | 8 | 17 | 0 | 14 | 5 | 3 | 3 | 0 | 0 | 9 | 13 | 0 | 3 | 0 | 17 | 8 |
| SCC-4 (*n* = 23) | 6 | 17 | 0 | 14 | 5 | 4 | 0 | 0 | 0 | 13 | 9 | 0 | 1 | 0 | 18 | 5 |
| SCC-5 (*n* = 10) | 3 | 7 | 0 | 9 | 0 | 1 | 0 | 0 | 1 | 1 | 8 | 0 | 0 | 0 | 5 | 5 |
| SCC-6 (*n* = 9) | 0 | 9 | 0 | 3 | 3 | 1 | 2 | 0 | 0 | 4 | 4 | 0 | 1 | 0 | 8 | 1 |

Then, we evaluated individually whether cancer stage, molecular classification, tumor differentiation grade, or overall survival status were associated with a glycogene signature; however, the results did not show any glycogene expression pattern. These data indicate that glycogene expression is not individually associated with any of the abovementioned clinical features. To refine the analysis, we exclusively analyzed SCC samples that contained complete information: cancer stage, molecular classification, tumor differentiation grade, and overall survival status (*n* = 158). The results showed that from 158 samples, 67 were organized into four clusters that displayed a specific glycogene expression pattern (Fig. 4A). The first cluster included 25 patient samples (SCC-3); the second cluster included 23 samples (SCC-4); the third cluster included ten samples (SCC-5); and the fourth cluster included nine samples (SCC-6) (Fig. 4A). Analysis of the clinical characteristics of clustered patient samples (Table 4) showed that SCC-3 tumors were stage I (56%), with a nonkeratinizing molecular classification (68%), a G3 differentiation grade (52%), and a 68% survival rate. Cluster SCC-5 was formed by samples from stage I (90%), with nonkeratinizing tumors in 70%, a G3 differentiation grade in 80% of the cases, and patients displayed a 50% survival rate. Concerning SCC-6, the cluster comprised only nonkeratinizing tumors with stages I to IV and an 88% survival rate (Table 4).

SCC-3 displayed a high expression of 6 glycogenes related to HS/heparin biosynthesis (Fig. 3B; Table S1-column D; Fig. S2C). SCC-5 showed a high expression of 9 glycogenes implicated in CS/DS biosynthesis and glycosphingolipid biosynthesis (lacto and neolacto series) (Fig. 4C; Table S1-column F; Fig. S2C). In contrast, SCC-6 exhibited a high expression of 42 glycogenes, some of which were involved in glycosphingolipid biosynthesis (ganglio and globo series) (Fig. 4E; Table S1-column G; Fig. S2C). Regarding SCC-4, the results showed two clusters of glycogenes, SCC-4A and SCC-4B, that corresponded to the same glycogenes in clusters SCC-1A and SCC-1B

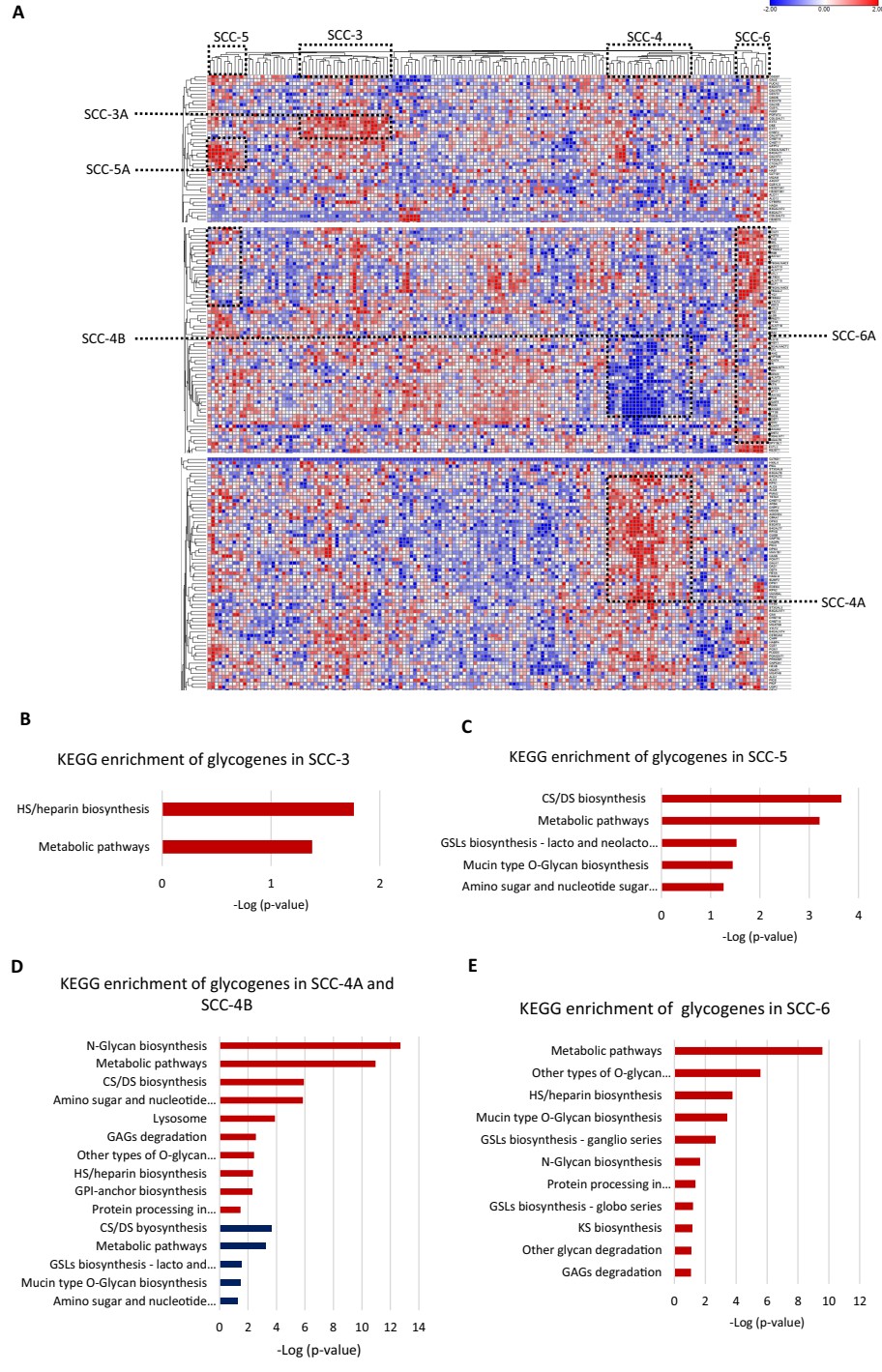

**Figure 4 Four clusters of cervical squamous carcinoma samples displayed a specific glycogene expression pattern.** (A) Heat map of glycogene expression of squamous carcinoma samples that contain complete clinical information. (B) KEGG enrichment analysis of glycogenes in SCC-3 and SCC-5 (C). (D) KEGG enrichment of glycogenes in SCC-4 with low expression in blue color and high expression in red color. (E) KEGG enrichment analysis in SCC-6. SCC: squamous carcinoma cluster; HS: heparan sulfate; GSLs: glycosphingolipids; CS: chondroitin sulfate; DS: dermatan sulfate; GPI: glycosylphosphatidylinositol; GAGs: glycosaminoglycans. Blue indicates low expression, and red indicates high expression.

(Tables S1-column E and S2-column F; Fig. S2A). A characteristic of SCC-4 was the low expression of genes involved in glycosphingolipid biosynthesis (lacto and neolacto series) and the high expression of genes involved in GAG degradation (Fig. 4D). In summary, all the data suggest that there are at least five types of SCC with distinguishable glycogene expression, where SCC-4 probably corresponds to SCC-1 since they share some samples and glycogenes (Table 5; Fig. S2B).

# DISCUSSION

## Cervical cancer displays upregulation of glycogenes implicated in GPI synthesis and downregulation of genes related to hyaluronan metabolism compared to healthy tissue

The human genome contains approximately 700 gene-encoding enzymes, including transporters and chaperones required for the cellular glycosylation machinery, glycan modifications, and degradation (Schjoldager et al., 2020). In this study, we analyzed the expression of 401 glycogenes, previously reported in Venkitachalam et al. (2016) and Aco-Tlachi et al. (2018), to obtain the glycome scenario in CC. We first compared the gene expression between CC and healthy tissue, and then we reached the glycogene expression in AC and SCC. The first analysis allowed us to understand the general glycosylation changes during cell transformation. The second analysis allowed us to identify the glycogene signatures in AC and SCC.

A comparison between CC and normal tissue indicated that CC displays upregulation of glycogenes related to GPI synthesis and downregulation of genes associated with the metabolism of HA. Regarding GPI-anchor biosynthesis, glycolipids act as anchors to specific cell surface proteins called GPI-anchored proteins. In breast, bladder, and gastric cancer, the high expression of other GPI-anchor biosynthesis glycogenes, such as *PIGU*, *PIGT*, and *PIGX*, is associated with oncogenesis, poor prognosis, and tumorigenesis (Guo et al., 2004; Zhao et al., 2012; Gamage & Hendrickson, 2013; Nakakido et al., 2016). Here, for the first time, we describe that in comparison with healthy tissue, CC also displays upregulation of two GPI-anchor biosynthesis glycogenes; interestingly, a high expression of some GPI-anchored proteins has been reported in CC (Jing et al., 2012; Liu et al., 2017; Jöhrens et al., 2019). To know whether the increase of GPI-anchored proteins is due to the rise of GPI biosynthesis should be tested in the future. Concerning HA, CC displayed downregulation of *HAS3* and *HYAL2* in comparison with healthy tissue. In HA metabolism, hyaluronan synthases 1 and 2 synthesize high molecular weight HA, while the hyaluronan synthase 3 (encoded by *HAS3*) synthesizes low molecular weight HA (Itano et al., 1999; Itano & Kimata, 2002); in contrast, HA degradation is induced by a family of hyaluronidases including Hyal2 (Csoka, Frost & Stern, 2001; Kaneiwa et al., 2012). Some evidence in SCC indicates that patients with a good prognosis display abundant levels of HA compared with those with the worst prognosis (Sano & Ueki, 1987). The accumulation of HA is typical in several types of cancers, and its levels directly correlate with increased malignancy and a poor prognosis (Passi et al., 2019). For example, in endometrial cancer, *HYAL1* and *HYAL2* are downregulated compared to healthy tissue,

Martinez-Morales et al.
2021
10.7717/peerj.12081

**Table 5 Glycogenes that are exclusively expressed in cervical adenocarcinoma and subtypes of squamous carcinoma.**

| Adenocarcinoma | Squamous carcinoma | | | |
|---|---|---|---|---|
| | SCC-1 and SCC-4 | SCC-3 | SCC-5 | SCC-6 |
| ALG11 | ALG3 | CHST2 | B4GALT1 | ARSB |
| ALG12 | ALG5 | COLGALT1 | C1GALT1 | B3GALNT1 |
| ALG9 | AMDHD2 | DSE | CHST11 | B3GNT2 |
| AMY1A | ARSA | EXT1 | CHST15 | B4GALT6 |
| AMY2A | B3GALT6 | EXT2 | CSGALNACT1 | CHSY3 |
| AMY2B | B3GAT3 | GALNT18 | GALNT2 | CTBS |
| ARSD | B4GALT2 | | GFPT2 | DPY19L1 |
| ARSE | B4GALT7 | | UAP1 | DSEL |
| B3GALT5 | CHPF2 | | | EXTL1 |
| B3GAT1 | CHST12 | | | EXTL2 |
| B3GNT3 | DAD1 | | | GALNT13 |
| B3GNT7 | DPM2 | | | GALNT15 |
| CHST4 | DPM3 | | | GALNT17 |
| CHST6 | EDEM2 | | | GCNT1 |
| CYB5R3 | GALK1 | | | GLCE |
| DPAGT1 | GNPTG | | | GLT8D2 |
| GAL3ST1 | GPAA1 | | | GNS |
| GAL3ST2 | GUSB | | | GXYLT1 |
| GALNT12 | HEXA | | | GXYLT2 |
| GALNT4 | HEXD | | | GYG1 |
| GALNT6 | MAN1B1 | | | HS2ST1 |
| GALNT9 | MANBAL | | | KL |
| GALNTL6 | MOGS | | | LFNG |
| GBA3 | NAGLU | | | MAN1A1 |
| GCNT2 | NAGPA | | | PIGK |
| GCNT3 | NANS | | | POFUT1 |
| GLB1L | NT5M | | | ST3GAL2 |
| GMDS | OST4 | | | ST6GAL2 |
| GNE | PIGQ | | | ST6GALNAC3 |
| GYG2 | PIGU | | | ST6GALNAC5 |
| MGAT3 | PMM1 | | | ST8SIA2 |
| NAGA | POMT1 | | | SULF1 |
| NEU4 | RFNG | | | TUSC3 |
| POFUT2 | RPN1 | | | UGGT1 |
| PYGB | RPN2 | | | UGGT2 |
| ST6GALNAC1 | ST3GAL3 | | | UXS1 |
| ST6GALNAC4 | SUMF2 | | | |
| ST6GALNAC6 | TSTA3 | | | |
| STT3A | | | | |
| SULT1C2 | | | | |
| TREH | | | | |
| UGT2B7 | | | | |

and this phenotype correlates with the accumulation of hyaluronan (*Nykopp et al., 2010*). Studies in animals indicate that *HYAL2* or *HYAL1* inhibits tumor growth and may control intercellular interactions (*Wang et al., 2008*). Examination of whether downregulation of *HYAL2* could lead to HA accumulation can be interesting in CC. Concerning *HAS3*, it is unknown whether the low expression of *HAS3* leads to a specific decrease in low molecular HA synthesis. Still, the low expression is associated with a poor prognosis in urothelial carcinoma (*Chang et al., 2015*). Characterization of the content levels of high-or low-molecular-weight HA in CC can be interesting, especially because their effects are different in cancer cell behavior and chemotherapy resistance (*Price, Lokman & Ricciardelli, 2018*; *Tavianatou et al., 2019a*; *Tavianatou et al., 2019b*). In summary, the results suggest that CC may display an increase in the synthesis of GPI anchors and dysregulation in the metabolism of HA.

The results obtained by the microarray assay allowed us to understand the global changes from the normal cervix to a cancer stage, and some of them show some coherence with the RNAseq analysis. Thus, the low expression of *UGT2B4* and *UGT2B28* in CC, analyzed in the microarray assay, was also found in all CC samples in the RNAseq analysis (Table S2-column B). Moreover, the low expression of *HAS3*, *HPSE*, and *GYS1* was found in the AC-B and AC-C clusters (Table S2-column C and column D, respectively). In the same way, the high expression of *ST6GAL1* was also found in the cluster AC-A (Table S1-column B). Despite these results, one limitation of our study is to validate the glycogene expression by RT-PCR. However, the validation can be difficult, since implies to have a previously characterized glycogene expression of the samples, considering the different glycogene signatures found in the SCC. Hence, the microarray assay of pooled samples allowed us to understand global changes of CC but not the molecular diversity of CC.

## Adenocarcinoma displays unique glycogene expression, but squamous cancer shows at least five types of glycogene signatures

Genomics studies suggest that CC is not a unique entity from molecular and genetic viewpoints (*The Cancer Genome Atlas Research Networkk (TCGA), 2017*; *Srinivasan, 2019*). Molecular characterization of CC includes three subgroups: keratin-low squamous, keratin-high squamous, and adenocarcinoma-rich (*The Cancer Genome Atlas Research Networkk (TCGA), 2017*). Regarding the glycome, our result suggests that AC displays a unique glycogene expression signature distinguishable from SCC, which in turn displays at least five types of glycogene signatures. Results shown in this study could explain some of the results where certain glycogenes in CC display high expression while other are exhibit low expression (*Varchalama et al., 2009*), and even where some of them are associated with specific cancer characteristics associated with patient prognosis (*Varchalama et al., 2009*; *Wang et al., 2003*).

First, we showed that AC displays a glycogene signature characterized by 42 glycogenes (Table 5; Table S3), where some of them, such as *ALG11*, *ALG9*, and *B3GALT5*, can be relevant for cancer patients (*Yu et al., 2020*; *Liu et al., 2021*; *Liao et al., 2021*). In comparison, SCC was characterized by a heterogenicity of glycogene expression that

can be distinguished by a selective glycogene expression (Fig. S2). The results suggest that the composition of characteristics such as tumor stage, differentiation grade, molecular characterization, and global patient survival comprise a global value related to the glycogene expression pattern. SCC-2 was characterized only by the low expression of glycogenes (Table S2-column G). Also, we identified four kinds of SCC. Type SCC-1/SCC-4 displayed, specifically, the expression of 38 glycogenes, while six glycogenes recognized SCC-3. In comparison, SCC-5 showed the expression of eight glycogenes, and SCC-6 could be distinguished by 36 glycogenes (Table 5; Table S3). The glycogene signatures of SCC-1/4, SCC-2, and SCC-6 were associated with 88–89% of overall survival. In contrast, the glycogene signatures of SCC-3 and SCC-5 displayed 68% and 50% of overall survival, respectively. These results suggest that glycogenes *CHST2*, *COLGALT1*, *DSE, EXT1, EXT2*, and *GALNT18* in SCC-3 can be helpful as a biomarker of prognostic. In addition, detecting high levels of *CHST11*, *CHST15*, and *GFPT2* can also be useful as prognostic biomarkers since their increased expression is associated with 50% of overall survival (in SCC-5). In comparison, their low expression is associated with 88–89% of overall survival (SCC-2). Further characterization of protein expression levels characterization and their association with survival will be necessary to consider some key glycogenes as valuable biomarkers.

Regarding the different expression of glycogenes in SCC, the whole molecular context can affect the gene expression in CC: miRNA, methylation, signaling pathway signatures, structural aberration (*Yang et al., 2009*; *Farkas et al., 2013*; *Vojta et al., 2016*; *The Cancer Genome Atlas Research Networkk (TCGA), 2017*) and the integrated HR-HPV genome (*The Cancer Genome Atlas Research Networkk (TCGA), 2017*; *Aco-Tlachi et al., 2018*; *Cisneros-Ramírez et al., 2020*; *Gagliardi et al., 2020*), which we did not explore.

## Glycogene expression diversity in CC and putative consequences in their glycosylation pathways

Glycosylation changes in cancer can be due to changes in the availability and abundance of the sugar nucleotide donors and cofactors, altered enzyme activity, and to dysregulation at the transcriptional level of the glycogenes (*Pinho & Reis, 2015*; *Schjoldager et al., 2020*). The evidence shown here suggests that CC may display a glycome diversity due to variability in the glycogene expression, whether AC or certain SC types. Regarding AC, results indicate that it is characterized by the increase of genes implicated in KS biosynthesis, N-glycan biosynthesis, and glycosphingolipids genes related to ganglio and globo series. In contrast, AC displayed low expression of glycogenes involved in the CS/DS and HS biosynthesis, GAG degradation, and GPI-anchor biosynthesis (Fig. 5). Regarding SC, KEGG analysis revealed that the SCC-2 displayed low expression of glycogenes implicated in the CHS biosynthesis, GAG degradation, and glycosphingolipids biosynthesis (Fig. 5).

In comparison, SCC-1/SCC-4 displayed an increase in the glycogenes implicated in HS/heparin biosynthesis and GAG degradation (Fig. 5). Type SCC-3 exhibited high expression of glycogenes implicated in HS/heparin biosynthesis (Fig. 5). SCC-6 was distinguished by the high expression glycogenes involved in several glycosylation pathways

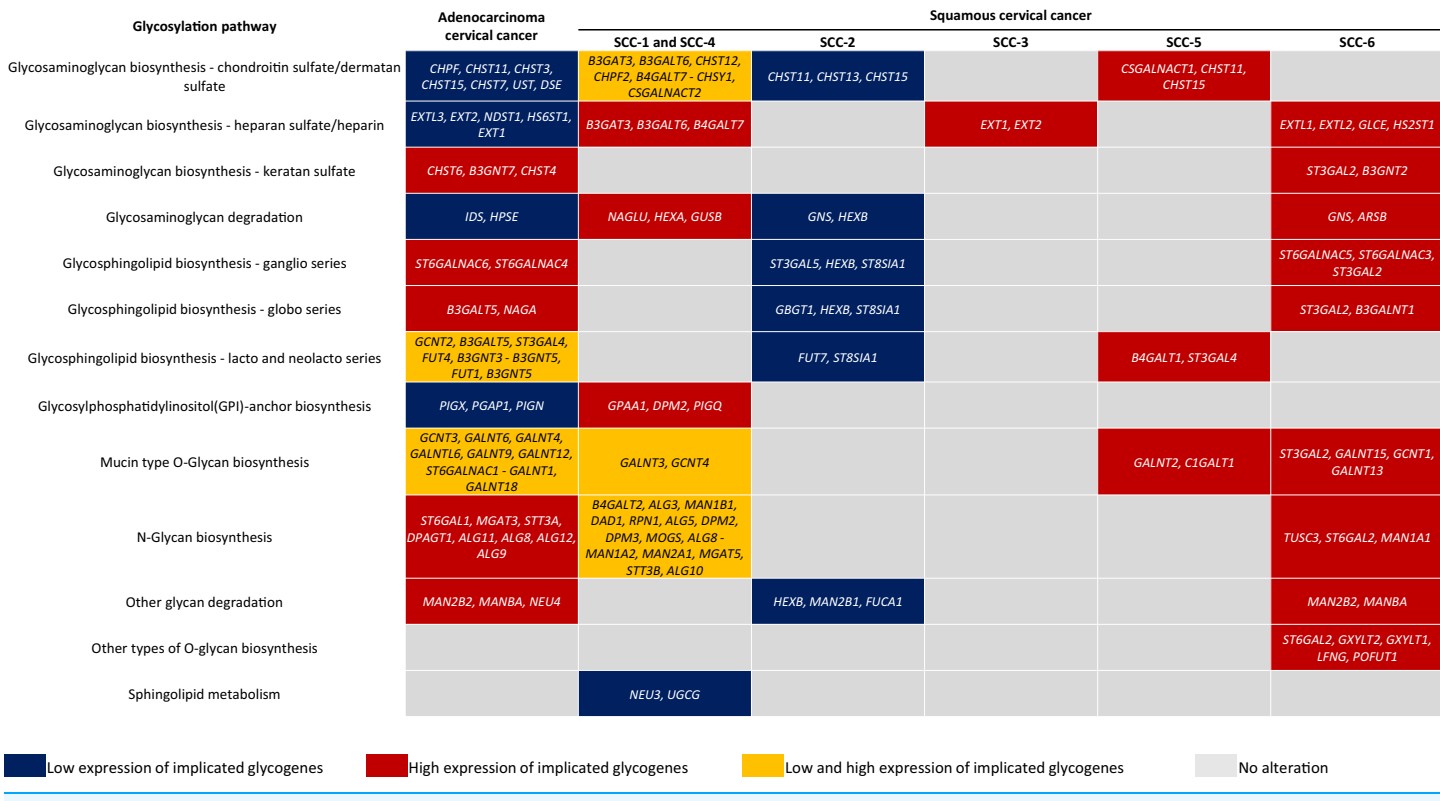

**Figure 5  Altered glycogenes in adenocarcinoma and types of squamous cancer and their respective implicated glycosylation pathways.** Cells in red indicate that all glycogenes display high expression compared to the rest of the cervical cancer samples, while cells in blue indicate that all glycogenes show low expression. Cells in yellow indicate that some of the glycogenes implicated in the same glycosylation pathway display high expression and others low expression; the glycogenes with high expression are located in the first part of the cell, while glycogenes with low expression are located in the second part and separated by a hyphen.

including in HS and KS biosynthesis and glycosphingolipid biosynthesis of the ganglio and globo series (Fig. 5). Finally, SCC-5 displayed the expression of eight glycogenes involved in CS/DS biosynthesis, glycosphingolipid biosynthesis of lacto and neolacto series, and mucin-type O-glycan biosynthesis (Fig. 5). All the data suggest a putative glycome diversity in CC, and future studies can be directed to characterize glycoconjugates accompanied by a glycogene signature.

# CONCLUSIONS

CC displays distinct molecular characteristics, and the evidence shown here suggests that the glycome can also be diverse. First, we showed that, in comparison with the cervix, CC displays upregulation of glycogenes involved in GPI synthesis and HA metabolism, suggesting a disruption in the type of molecular weight of HA. In addition, our results showed that AC displays a unique glycogene signature independent of SCC. Interestingly, adenosquamous carcinoma displayed the same signature as AC. In comparison, SCC displays a diversity of glycogene expression, and at least five types of SCC displayed unique glycogene signatures that can be distinguished from each other through a set of specific glycogenes. Notably, some types of SCC can be associated with certain tumor characteristics and patient survival.

Further analysis of the CC glycome will be necessary: First, to discern a possible association between glycogene expression and a value of the clinical characteristics of the tumors. Second, to clarify the molecular context that leads to the specific glycogene expression inside the CC population. Third, to confirm the putative glycan content changes in each case; and fourth, to know whether a glycogene signature or their encoded proteins can be used as a putative biomarker or a glycome-based classification. In summary, the differences between the AC and SCC types shown here indicate that CC should be deeply characterized to identify subtypes of CC that allow the development of targeted therapies according to the tumor characteristics.

## ACKNOWLEDGEMENTS

We thank Dr. Lorena Chávez González, Dr. Simón Guzmán León, Dr. José Luis Santillán Torres, and Dr. Jorge Ramírez for the technical assistance with the microarray determinations and Mr. Gerardo Coello, Mr. Gustavo Corral and Ms. Ana Patricia Gómez for their assistance with the genArise software.

### Funding

This work was supported by the Consejo Nacional de Ciencia y Tecnología (SALUD-2012-01-180219 and Catedras Program no. 485) and Instituto Mexicano del Seguro Social no. FIS/IMSS/PROT/G14/1293. The funders had no role in study design, data collection and analysis, decision to publish, or preparation of the manuscript.

### Grant Disclosures

The following grant information was disclosed by the authors:
Consejo Nacional de Ciencia y Tecnología: SALUD-2012-01-180219.
Catedras: 485.
Instituto Mexicano del Seguro: FIS/IMSS/PROT/G14/1293.

### Competing Interests

The authors declare that they have no competing interests.

### Author Contributions

- Patricia Martinez-Morales conceived and designed the experiments, performed the experiments, analyzed the data, prepared figures and/or tables, and approved the final draft.
- Irene Morán Cruz performed the experiments, authored or reviewed drafts of the paper, and approved the final draft.
- Lorena Roa-de la Cruz performed the experiments, authored or reviewed drafts of the paper, and approved the final draft.
- Paola Maycotte performed the experiments, analyzed the data, authored or reviewed drafts of the paper, and approved the final draft.
- Juan Salvador Reyes Salinas analyzed the data, authored or reviewed drafts of the paper, and approved the final draft.
- Victor Javier Vazquez Zamora analyzed the data, authored or reviewed drafts of the paper, and approved the final draft.
- Claudia Teresita Gutierrez Quiroz analyzed the data, authored or reviewed drafts of the paper, and approved the final draft.
- Alvaro Jose Montiel-Jarquin analyzed the data, authored or reviewed drafts of the paper, and approved the final draft.
- Verónica Vallejo-Ruiz conceived and designed the experiments, analyzed the data, prepared figures and/or tables, and approved the final draft.

## Human Ethics

The following information was supplied relating to ethical approvals (*i.e.*, approving body and any reference numbers):

This research was approved by the Ethics Committee of the Instituto Mexicano del Seguro Social, registration number R-2012-785-061.

## Data Availability

Data are available at NCBI GEO: GSE159976.

## Supplemental Information

Supplemental information for this article can be found online at http://dx.doi.org/10.7717/peerj.12081#supplemental-information.

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
