# Peer review of "Hallmarks of glycogene expression and glycosylation pathways in squamous and adenocarcinoma cervical cancer"

_PeerJ, doi:10.7717/peerj.12081_

## Round 0.1 · original submission · Major Revisions

Thank you for submitting your work to PeerJ. Your manuscript has been peer-reviewed. It is interesting. However, you must carefully revise your manuscript per the reviewers' comments.

Reviewer 1 ·

Basic reporting

.

Experimental design

.

Validity of the findings

.

Additional comments

Re: #60970
This manuscript claims that a comprehensive analysis of glycogene expression in cervical cancer tissues by the microarray expression analysis provides a possible prognostic biomarker for diagnosis of cervical carcinoma, i.e. metastatic potential, immune evasion and drug resistance. Although further reverification on the number of cases is necessary, the data presented are useful, and the manuscript is acceptable for publication.
As indicated by authors, glycogene expression is not directly correlated with final expression of glycoconjugates, and accordingly with cell biological properties of cervical carcinoma. The synthesis of glycoconjugates, i.e. GPI anchor, proteoglycan, glycosphingolipids, and N- and O-linked glycoproteins, is dependent on the supplies of substrate glycoconjugates and nucleotides sugars, and the enzymatic activities of glycosyltransferases and glycosidases. In addition, the cancerous tissues are the mixtures of cells with the different proliferation- and differentiation-stages, connective tissues, blood vessels and so on. Same as true for normal cervical tissues. Based on the present data, further evaluation on the expression of glycoconjugates and the characterization of cervical carcinoma is required.

Reviewer 2 ·

Basic reporting

The manuscript by Mertinez-Morales et al. investigated the glycogene expression and glycosylation pathways in squamous and adenocarcinoma cervical cancer using microarrays and RNA-Seq data. The purpose of the study is rather interesting, since glycogenes still remain one of the under explored area as compared to other genes. However, there are some drawbacks in the designing, data analyses and interpretation of the results.

The introduction lacks a proper definition of glycogenes. In addition to the two enzymes mentioned by the authors on line 67, other types of genes are also classified as glycogenes (https://doi.org/10.1186/1471-2164-10483). It should be highlighted if authors want to focus on any specific set of glycogenes.

Line 111: Please provide mean age and standard deviation for both samples.

The discussion is extremely long which makes the manuscript a difficult read.

Experimental design

It is not clear what was the aim of carrying out microarray analysis. Bulk of their results and discussion is based on the analysis of publicly available RNA-Seq datasets. Besides, the microarray platform used consists of only 10,000 genes, and therefore may comprise only a limited number of glycogenes. This might explain why only 15 glycogenes were identified as differentially expressed in their microarray dataset. It would be interesting to know how many were actually glycogenes from the list of 10,000 genes.

Additionally, why did authors chose Microarrays over the RNA-Seq?

The authors must mention the version number of databases and tools for reproducibility. For example, using the list of 6 up-regulated glycogenes I could not find any interaction between these 6 genes (interaction score = 0.9). It would have been better if the interaction network was generated using 401 glycogenes and then map the 15 genes identified from microarray onto that network.

What fold change is represented by the 15 altered glycogenes? Moreover, the authors did not validate by experimental methods (e.g. RT-PCR) any of the 15 altered glycogenes.The authors should also improve on the presentation of protein-protein and pathway enrichment analyses.

Validity of the findings

It is also not discussed how many out of their 15 glycogenes were also identified in RNA-Seq analysis. Reasons for any discrepancies should be discussed. Please report adjusted p-values for enrichment analyses.

Line 220: Table Supp 2 is cited before Table Supp 1.

Figure 2 and 3: Since the genes belong to different clusters in these figures, it is suggested authors highlight these genes (probably by different colors) in supp. Tables 1 and 2. It is difficult to follow which genes from supplementary data belong to which clusters.

Line 265: impllicated -> implicated.

Information in Table 5 and supplementary tables is limited. It is recommended to include the gene/protein names and their functional annotations.

Additional comments

The authors should look at some of the representative papers focusing on glycogene (
https://doi.org/10.1371/journal.pone.0013002; https://doi.org/10.1186/1471-2164-10-483) analyses as well as papers with PPI network and enrichment analyses (https://doi.org/10.1038/s41598-020-65909-x). This will help them to improve the manuscript substantially by properly analyzing the gene expression data.

Reviewer 3 ·

Basic reporting

The article needs revison for English langaguge, not thourgh though.

Introduction and Discussion are very long and need extensive revision.

Experimental design

Needs more studies to include.

Exclusiion and inclusion criteria of subjects are poorly defined.

Validity of the findings

Hypothesis and experimental design is good, however, there is so much repetition of results. Needs extensive revision and cutting down the introduction and discussion.

Additional comments

The article needs revison for English langaguge, not thourgh though.

---

## Round 0.2 · accepted · Accept

I am writing to inform you that your manuscript - Hallmarks of glycogene expression and glycosylation pathways in squamous and adenocarcinoma cervical cancer - has been Accepted for publication. Congratulations!

Reviewer 2 ·

Basic reporting

The manuscript looks improved now and seems acceptable for publication.
However, please add a legend for the supplementary data.

Experimental design

No comment

Validity of the findings

No comment